# Language Pretraining Gives Structured Forecasters a Sequential Prior

**Zhenghan Tai** [1*]  **Alexis Roger** [2 3*]  **Prateek Humane** [4 3*]
**Gwen Legate** [5 3]  **Andrei Mircea** [4 3]  **Vasilii Feofanov** [6]  **Irina Rish** [4 3]

## Abstract

Structured-data foundation models now target transfer across time series, tabular data, text, and other sequential domains, but it remains unclear what language pretraining contributes to forecasting beyond a generic transformer architecture. We study this question by adapting Qwen3-0.6B to probabilistic time-series forecasting on GiftEval and comparing language-pretrained models with identical randomly initialized models under full and parameter-efficient finetuning. Language pretraining gives a large early optimization advantage, especially in limited-adaptation regimes: LoRA attention updates recover most of the effective transfer benefit of full finetuning. Frozen-state probes, retrieval forecasts, gradient coherence, and effective-rank dynamics indicate that language models already contain reusable sequential structure before time-series supervision. These results frame language-to-time-series transfer as efficient low-rank specialization of a pretrained sequential inductive bias, with implications for future structured-data foundation models.

## 1. Introduction

Foundation models for structured data are moving from single-domain pretraining toward systems that can reuse structure across time series, tables, event streams, and text. Time-series forecasting is a useful benchmark for this goal: it is a core structured-data task, it stresses sequential prediction under distribution shift, and it exposes whether transfer comes from reusable temporal structure rather than from memorized linguistic semantics.

Recent work has adapted language models to forecasting through prompting, reprogramming, tokenized forecasting, and parameter-efficient finetuning (Gruver et al., 2023; Jin et al., 2023; Zhou et al., 2023; Ansari et al., 2024; Wolff et al., 2025). The evidence is mixed. Some studies argue that architecture and tokenization explain much of the benefit (Tan et al., 2024; Zheng et al., 2025; Zhang et al., 2025); others find that language initialization helps most in low-data, cross-domain, or distribution-shift settings (Riachi et al., 2025; Qiu et al., 2026). This tension is central for structured-data foundation models: when should we reuse a pretrained language backbone, and what transferable structure does it provide?

These empirical tensions connect to a broader theoretical question about cross-modal structure. The Platonic Representation Hypothesis (Huh et al., 2024) suggests that sufficiently scaled models converge toward a shared statistical structure regardless of modality, and frozen vision backbones have been shown to forecast time series (Chen et al., 2024). Meanwhile, work on intrinsic dimensionality (Aghajanyan et al., 2021) shows that pretrained models can be finetuned within low-dimensional subspaces, and zero-sum learning dynamics (Mircea et al., 2025) reveal that per-example gradient conflicts can bottleneck training from scratch. Taken together, language pretraining supplies both a compatible representation geometry and a favorable optimization landscape for time-series adaptation

We test this hypothesis empirically. Our thesis is that autoregressive language pretraining creates a reusable sequential inductive bias that is useful for forecasting, and that adaptation behaves primarily as low-rank specialization rather than full relearning. Language contains repetition, drift, local discontinuities, long-range continuation, and multi-scale dependencies; these are not time-series semantics, but they are sequential structures that forecasting systems also need. Under this view, language pretraining supplies candidate directions, while time-series finetuning selects and reshapes them for numerical prediction.

Our contributions are:

- We show forecasting transfer from language-pretrained Qwen3-0.6B to GiftEval, with the strongest gains early in training and under limited adaptation.

---

[1]University of Toronto [2]McGill University [3]Mila - Quebec AI Institute [4]Université de Montréal [5]Concordia University [6]42.com. Correspondence to: <alexis.roger@mila.quebec>.

*Proceedings of the 2nd ICML Workshop on Foundation Models for Structured Data*, Seoul, South Korea. 2026. Copyright 2026 by the author(s).

- We find that transfer behaves as low-rank specialization: attention LoRA recovers most of the effective data-transfer advantage of full finetuning.

- We analyze optimization geometry and show that pretrained models begin time-series training with coherent per-example gradients, unlike random initialization.

- We provide evidence that forecasting-relevant temporal structure exists before finetuning through frozen linear probes and retrieval-based forecasts.

## 2. Structured Forecasting Setup

We repurpose Qwen3-0.6B (Yang et al., 2025) as a probabilistic time-series forecaster by casting continuous forecasting as next-token prediction over discretized values. For each univariate series, we sample windows with context length $C = 512$ and forecast horizon $L = 64$. Context statistics are used for per-series normalization, and values are discretized into $V = 1024$ uniform bins over $[-5, 5]$. Bin indices are mapped directly to the first vocabulary positions, while Qwen special tokens are preserved, following the small-vocabulary tokenization strategy used in recent time-series language models (Roger et al., 2025).

The model is trained with quantile loss over $\mathcal{Q} = \{0.1, \ldots, 0.9\}$:

$$\mathcal{L} = \frac{1}{T|\mathcal{Q}|} \sum_{t=1}^{T} \sum_{\tau \in \mathcal{Q}} \rho_\tau(y_t - \hat{q}_{t,\tau}),$$
$$\rho_\tau(u) = u(\tau - \mathbf{1}[u < 0]). \quad (1)$$

Output logits define a categorical distribution over ordered bins; inverting its CDF yields non-crossing probabilistic forecasts without repeated forward passes.

We train on sliding windows from GiftEval (Aksu et al., 2024) and evaluate on held-out windows using CRPS, MASE, and MSE under GluonTS-style aggregation (Alexandrov et al., 2020). We compare a language-pretrained initialization with an identical randomly initialized Qwen3 architecture. For the pretrained model, we consider four adaptation regimes: full finetuning, IO-only tuning of embeddings and output head, rank-8 LoRA on attention projections, and LoRA attention plus trainable IO.

## 3. Transfer Results

Figure 1 shows the main training dynamics on a held-out GiftEval solar/10T slice. Across CRPS, MASE, and MSE, language-pretrained models begin improving earlier than randomly initialized counterparts. The gap is most important in the low-step regime, where structured-data adaptation is practically constrained by data, compute, or rapid deployment requirements.

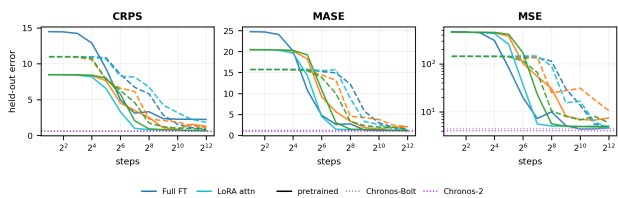

*Figure 1.* **Training dynamics on held-out GiftEval solar/10T.** Solid lines use language-pretrained Qwen3-0.6B; dashed lines use the same architecture from random initialization. Language pretraining gives an early forecasting-transfer advantage across full finetuning, IO-only tuning, and LoRA adaptation.

*Table 1.* Early forecasting transfer at step 128 on held-out GiftEval. $D_T$ is the effective data-transfer multiplier: how many more steps the random model needs to match the pretrained model at a target CRPS.

| Regime | PT CRPS | RI CRPS | PT MASE | RI MASE | $D_T$ |
|---|---|---|---|---|---|
| Full FT | 49.51 | 151.8 | 1.56 | 7.82 | 6.8 |
| IO only | 59.56 | 111.6 | 3.43 | 6.57 | 2.3 |
| LoRA attn | **20.10** | 154.5 | 1.61 | 7.87 | 5.2 |
| LoRA attn+IO | 22.54 | **50.68** | 1.70 | **3.44** | 3.4 |

The early checkpoint results in Table 1 show the same pattern on the broader held-out evaluation. At step 128, pretrained LoRA attention reaches CRPS 20.10 while the corresponding random initialization remains at 154.5; pretrained LoRA+IO reaches 22.54 versus 50.68. Strong specialized forecasters still perform best at this point (Chronos-2 reaches CRPS 10.89 and ARIMA 11.94; see Appendix A.4), but the comparison of identical backbones isolates the transfer effect. A zero-shot text Qwen3 baseline performs orders of magnitude worse, which confirms that the language prior helps only after structured-data adaptation.

The central practical result is that language pretraining improves forecasting transfer most when adaptation is limited. Full finetuning has the largest effective-transfer multiplier, but attention-only LoRA preserves most of it. Parameter-efficient finetuning therefore matches the structure of the transfer problem rather than approximating full FT.

## 4. Why Transfer Occurs

**Temporal structure before finetuning.** We first ask whether frozen language states are already compatible with real time series. We pass 1,920 WikiText-103 sequences of length 512 through frozen Qwen3-0.6B, concatenate hidden states from all 28 layers,

$$h_{i,t} = [h_{i,t}^{(0)}; \ldots; h_{i,t}^{(27)}] \in \mathbb{R}^{28672}, \quad (2)$$

and train a single linear map $\hat{y}_{i,t} = w^\top h_{i,t} + b$ to produce scalar trajectories. There is no paired text–time-series supervision. Instead, an EM-style nearest-neighbor objective matches projected trajectories to a bank of 10,000 z-scored

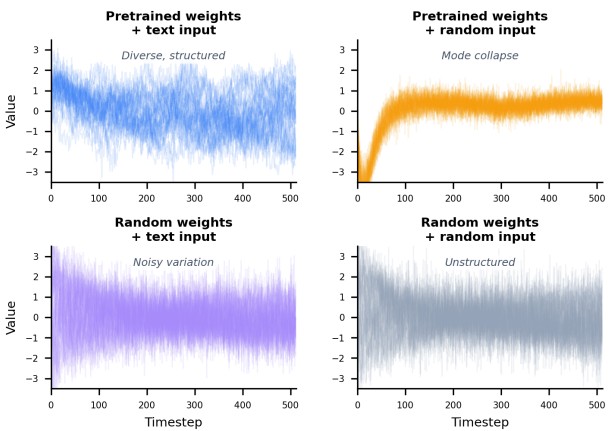

*Figure 2.* **What creates temporal structure in frozen LLM states?** $2 \times 2$ ablation crossing model weights (pretrained vs. random) with input (text vs. random tokens). Only the combination of pretrained weights and meaningful text input produces diverse, realistic temporal trajectories.

GiftEval windows, with a spectral diversity penalty to avoid collapse.

The projection produces diverse realistic trajectories: nearest neighbors span 686 distinct real series, and held-out WikiText projections have similar nearest-neighbor MSE to training text (0.72 vs. 0.67). A $2 \times 2$ ablation (Figure 2) separates architecture, input, and pretraining. Random weights with text produce noisy variation; pretrained weights with random tokens collapse to a few shapes; only pretrained weights traversed by natural text produce diverse temporal trajectories. This suggests that useful multimodal sequential structure is present before forecasting supervision.

**Retrieval forecasts from projected states.** The frozen projections are not just shape matches. Given the first half of 500 real GiftEval windows, we retrieve the closest projected WikiText trajectory from a bank of 10,000 projections and use its second half as the forecast. This retrieval forecast achieves MSE 1.91 versus 2.27 for last-value carry-forward, a 16% reduction. It wins only 37% of cases, but when it wins the median improvement is roughly $4\times$, typically on trends and level shifts. Forecasting-relevant continuation signal exists in frozen language representations, without any paired supervision.

**Optimization geometry.** We compute per-example gradients $g_i = \nabla_\theta \mathcal{L}(x_i; \theta)$ for a fixed held-out batch of 32 time-series examples and track mean off-diagonal cosine similarity,

$$A(\theta) = \frac{1}{N(N-1)} \sum_{i \neq j} \cos(g_i, g_j). \quad (3)$$

Figure 3 illustrates this on a sine input: LangInit inherits

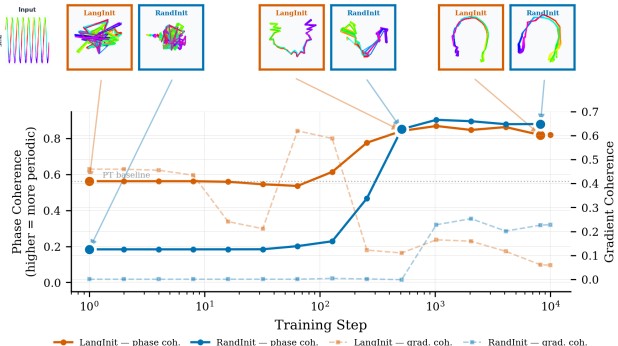

*Figure 3.* **Phase and gradient coherence on a sine input.** LangInit (top) inherits periodic structure and starts coherent; RandInit (bottom) begins near zero and undergoes a phase transition around step 300. t-SNE insets at steps 1/512/8192 show the corresponding hidden-state reorganization.

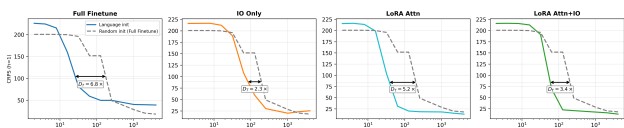

*Figure 4.* **Effective data transfer per regime.** CRPS (h=1) vs. training step; solid = language-pretrained, dashed = random-init full-FT reference. $D_T$ is the step-multiplier RandInit needs to match LangInit at a target CRPS. LoRA-attention preserves most of full-FT's transfer advantage.

periodic structure from pretraining and starts with high coherence, while RandInit begins near zero and undergoes an abrupt phase transition around step 300 where gradient coherence spikes as the model discovers periodic geometry. This coupling between coherence onset and loss descent holds across all four adaptation regimes, including IO-only where the backbone is frozen. The pattern connects to zero-sum learning (Mircea et al., 2025): randomly initialized models must escape a phase of destructive gradient interference before optimization can progress, while pretrained models bypass it entirely.

## 5. Optimization and Low-Rank Specialization

Our LoRA (Hu et al., 2021) results imply a low-dimensional adaptation path. Figure 4 reports the effective data-transfer multiplier $D_T$ (Hernandez et al., 2021) per regime: full FT $D_T = 6.8$, attention-only LoRA $D_T = 5.2$, LoRA+IO $D_T = 3.4$, IO-only $D_T = 2.3$. In each panel, the horizontal gap between the pretrained and random-init CRPS curves directly shows how many more training steps random initialization needs to match the pretrained model at a given loss level. A rank-8 perturbation to attention projections captures most of the language-to-time-series advantage.

Effective-rank dynamics give a representation-level view. For each checkpoint and transformer layer, we collect hid-

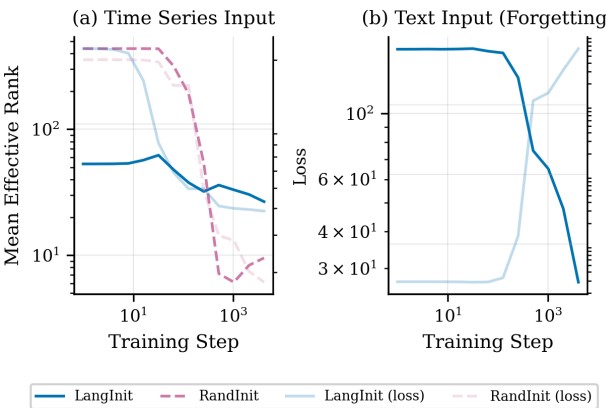

*Figure 5.* **Effective rank across training.** Left: on time-series input, RandInit starts near-isotropic (∼440) and collapses to ∼10; LangInit declines gradually from ∼50 to ∼27. Right: on text input, LangInit's effective rank drops 85% while perplexity rises, confirming catastrophic forgetting. Transparent lines show CRPS on a secondary axis.

den states from 10,000 validation windows and compute $\text{erank}(\Sigma) = \exp(-\sum_i p_i \log p_i)$, where $p_i$ are normalized covariance eigenvalues (Roy & Vetterli, 2007). As shown in Figure 5, random initialization begins near-isotropic on time-series inputs, with mean effective rank around 440, then collapses to about 10 within 500 steps. Language initialization starts already compressed, around rank 50, and declines more gradually to about 27. On text inputs, finetuning causes an 85% reduction in effective rank (from ∼165 to ∼25), confirming catastrophic forgetting of language-specific directions.

The compression is not uniform across layers. Per-layer analysis shows that middle layers (∼8–17) *increase* effective rank during finetuning while early and late layers contract—finetuning redistributes capacity from the network's periphery to its core. This is consistent with the representation phases described by Li et al. (2025): the randomly initialized model must first undergo a warmup phase of representational collapse before rebuilding, while the language-initialized model skips this phase and moves directly toward selective compression, preserving variance along directions useful for forecasting.

**Reuse vs. reinvention.** To compare how the two initialization paths represent temporal structure, we pass synthetic waveforms (sine, square, sawtooth) through trained models and visualize hidden-state geometry with 2D PCA at layer 13. RandInit consistently produces clean, low-dimensional arcs with 62–92% of variance in two components. LangInit produces geometrically richer trajectories with only 25–41% PCA variance, distributing periodic structure across a higher-dimensional inherited representation. Despite these geometric differences, centered kernel alignment (CKA) between the two models' layer-13 rep-

*Table 2.* Shared pretrained–finetuned crosscoder features. Shared features concentrate in layers 7–10, linking time-series motifs to structurally analogous text contexts.

| Lyr | Feat | TS motif | Text motif |
|---|---|---|---|
| 10 | 1712 | magnitude jumps | measurements / units |
| 9 | 2469 | volatile regimes | cyclone narratives |
| 7 | 3888 | isolated spikes | timestamped events |
| 8 | 2567 | missing windows | incomplete refs |

resentations is 0.855–0.988 across waveforms, which confirms that they recover nearly the same relational geometry up to rotation and scaling. The models arrive at functionally equivalent temporal representations through different paths: RandInit builds compact structure from scratch, while LangInit reshapes a richer pretrained manifold.

**Compact interpretability check.** We use sparse crosscoders only as a diagnostic. Sparse crosscoders (Lindsey et al., 2024) trained between pretrained and finetuned activations identify shared features concentrated in layers 7–10. Table 2 shows representative examples: the strongest shared features link time-series motifs to text contexts with analogous sequential structure. This does not imply semantic equivalence between text and time series. It supports the weaker and more relevant FMSD claim: language pretraining supplies reusable sequential primitives that can be specialized for structured forecasting.

## 6. Implications for Structured-Data FMs

**Alexis Roger**[2 3] **Prateek Humane**[4 3] **Zhenghan Tai**[1]
**Gwen Legate**[5 3] **Andrei Mircea**[4 3] **Vasilii Feofanov**[6]
**Irina Rish**[4 3]

Cross-domain transfer matters most when adaptation is limited or under distribution shift. Language-pretrained backbones provide a sequential prior that dedicated training needs many examples to discover—not linguistic knowledge, but reusable sequential structure. Because adaptation operates as direction selection, parameter-efficient methods are a structural match, not just a cheaper approximation. This introduces a scaling dimension orthogonal to dataset size: *schema diversity through cross-domain pretraining*.

**Limitations.** All experiments use one backbone (Qwen3-0.6B), one tokenization scheme, and univariate forecasting. The linear probe shows representation compatibility, not optimal zero-shot forecasting. Effective-rank and gradient analyses support low-rank specialization but do not fully disentangle geometry from initialization statistics.

**Conclusion.** Language pretraining helps structured forecasting by supplying reusable sequential structure; finetuning specializes it via low-rank adaptation. Structured-data foundation models can inherit useful inductive biases from

sequential pretraining and specialize them under rigorous evaluation.

## 7. Acknowledgment

We thank 42.com for providing computational resources that supported this work. We also acknowledge the AMD University Program AI & HPC Cluster for additional compute resources used in this research.

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

# A. Experimental Setup

## A.1. Hyperparameters and Reproducibility

Table 3 summarizes all hyperparameters used across our experiments; unless otherwise noted, all runs share this configuration.

## A.2. Evaluation Metrics

This section details the evaluation metrics implemented for assessing the forecasting performance of the models. For all fixed-point metrics, the 0.5 quantile (median) is extracted from the probabilistic forecasts and used as the deterministic prediction.

**Aggregation Strategy and Implementation:** In our evaluation methodology, we follow closely the GluonTS library (Alexandrov et al., 2020), particularly, the aggregation approach. All metrics described below are computed *per-sequence first*, and then averaged across all sequences in the dataset. Specifically, the error or score is aggregated over the forecast horizon $H$ for each individual time series, and the final reported metric is the unweighted mean of these per-sequence scores. This macro-averaging ensures that every sequence contributes equally to the final evaluation, regardless of its absolute magnitude or scale.

To ensure robust evaluation, any time steps containing NaN values in the ground truth are dynamically excluded from computation by masking. Additionally, a small utility constant ($\epsilon = $ 1e-8) is added to denominators in percentage and scale-normalized metrics to prevent division by zero.

### A.2.1. POINT FORECAST METRICS

**Basic Error Metrics** Mean Squared Error (MSE) quantifies prediction accuracy by heavily penalizing larger errors, making it sensitive to outliers. Root Mean Squared Error (RMSE) provides the error in the original data units while retaining MSE's sensitivity.

$$\text{MSE} = \frac{1}{H} \sum_{t=1}^{H} (y_t - \hat{y}_t)^2$$

$$\text{RMSE} = \sqrt{\text{MSE}}$$

where $H$ is the forecast horizon, $y_t$ is the true value, and $\hat{y}_t$ is the predicted median.

Mean Absolute Error (MAE) applies a linear penalty to measure the average absolute difference:

$$\text{MAE} = \frac{1}{H} \sum_{t=1}^{H} |y_t - \hat{y}_t|$$

**Normalized Metrics** Mean Absolute Scaled Error (MASE) measures forecast accuracy relative to a naive baseline, allowing for cross-series comparison across data with vastly different scales.

$$\text{MASE} = \frac{\text{MAE}}{\frac{1}{H-m}\sum_{t=m+1}^{H}|y_t - y_{t-m}|}$$

The seasonality parameter $m$ is selected based on the expected periodic patterns of the timeframe ($m = 1$ for a general naive baseline, $m = 60$ for 1-minute data, and $m = 24$ for 1-hour data).

To further achieve scale-invariance, RMSE and absolute deviations are normalized by the scale of the data (mean or sum of absolute values) to compute the Normalized Root Mean Squared Error (NRMSE) and Normalized Deviation (ND):

$$\text{NRMSE} = \frac{\sqrt{\frac{1}{H}\sum_{t=1}^{H}(y_t - \hat{y}_t)^2}}{\frac{1}{H}\sum_{t=1}^{H}|y_t|}$$

$$\text{ND} = \frac{\sum_{t=1}^{H}|y_t - \hat{y}_t|}{\sum_{t=1}^{H}|y_t|}$$

**Percentage-Based Metrics** Mean Absolute Percentage Error (MAPE) provides a scale-independent metric as a percentage. To address MAPE's asymmetry problem and instability near zero, the Symmetric Mean Absolute Percentage Error (sMAPE) equally penalizes over-predictions and under-predictions:

$$\text{MAPE} = \frac{100}{H}\sum_{t=1}^{H}\frac{|y_t - \hat{y}_t|}{|y_t| + \epsilon}$$

$$\text{sMAPE} = \frac{200}{H}\sum_{t=1}^{H}\frac{|y_t - \hat{y}_t|}{|y_t| + |\hat{y}_t| + \epsilon}$$

### A.2.2. PROBABILISTIC METRICS

**Continuous Ranked Probability Score (CRPS)** The approximated CRPS measures the accuracy of probabilistic forecasts by evaluating the entire predictive distribution using quantile loss (QL).

$$\text{CRPS}_{\text{approx}} = 2\sum_q w_q \cdot \text{QL}_q(y_{\text{true}}, y_{\text{pred}})$$

where $\text{QL}_q(y, \hat{y}) = (q - \mathbf{1}_{y \le \hat{y}}) \cdot (y - \hat{y})$ and $w_q$ represents the discrete weight calculated based on the distance between quantile levels.

**Weighted Mean Absolute Percentage Quantile Loss (wMAPE)** To compare probabilistic performance across heterogeneous time series, we implement a scale-invariant version of quantile loss weighted by the sum of absolute true values:

$$\text{wMAPE} = \frac{1}{Q}\sum_{q=1}^{Q}\frac{\sum_{t=1}^{H}\text{QL}_q(y_t, \hat{y}_{q,t})}{\sum_{t=1}^{H}|y_t|}$$

### A.2.3. DIRECTIONAL AND CORRELATION METRICS

**Directional Accuracy (DA)** DA evaluates how often the model correctly predicts the direction of movement relative to a historical anchor point $y_0$ (the last observed value).

$$\text{DA} = \frac{1}{H}\sum_{t=1}^{H}\mathbf{1}_{\text{sign}_\tau(\hat{y}_t - y_0) = \text{sign}_\tau(y_t - y_0)}$$

**Correlation Metrics** To evaluate the model's ability to capture trend dynamics independently of absolute magnitude errors, we utilize Pearson correlation coefficients. Pearson evaluates the linear relationship between predictions and ground truth.

## A.3. Experimental Results

Here we provide the complete set of evaluation results across all training regimes and metrics. Figure 6 shows the training progression of all eleven h=1 metrics over 4096 training steps, revealing a consistent pattern across metrics: language-pretrained models begin converging within the first 10–100 steps, while randomly initialized models require half an order of magnitude more training to reach comparable performance. By step 4096, both initialization strategies converge to similar error levels across all regimes, confirming that the pretrained weights accelerate optimization rather than alter the final solution. Tables 4 and 5 report the full numerical results at step 128 for single-step (h=1) and multi-step (h=64) forecasting respectively, alongside four established baselines and a zero-shot Qwen3 text baseline. At this early checkpoint, the transfer advantage of language pretraining is clearly visible: pretrained LoRA Attn achieves a CRPS of 20.10 compared to 154.5 for its randomly initialized counterpart, while all pretrained regimes outperform the random initializations that have not yet begun to converge. The Qwen3 text baseline performs orders of magnitude worse than all trained models, confirming that the language model's pretrained representations require domain-specific finetuning to be useful for time series forecasting.

## A.4. Effective Transfer

Let $\mathcal{L}_R(d)$ and $\mathcal{L}_P(d)$ denote the validation losses achieved after training for $d$ time series tokens using random and pre-trained initializations, respectively. For a given validation loss $\ell$, $\mathcal{L}_\cdot^{-1}(\ell)$ therefore denotes the amount of data required to train a model to reach the loss level $\ell$ begining from a given initialization. We define transfer as "effective" or "positive" when starting the time-series model training with the language model's weights allows us to achieve the same validation loss with less data than a model initialised randomly. This data quantity difference is what we refer to as the amount of data we "saved". Concretely, the *effective data transferred* $D_T(\ell)$ for a target loss $\ell$ is defined as the difference of these data amounts:

$$D_T(\ell) := \mathcal{L}_R^{-1}(\ell) - L_P^{-1}(\ell).$$

A positive $D_T(\ell)$ indicates that initializing the model from pretrained language weights requires fewer training examples to reach the loss $\ell$ compared to random initialization, signifying an effective transfer from the upstream task to time series forecasting.

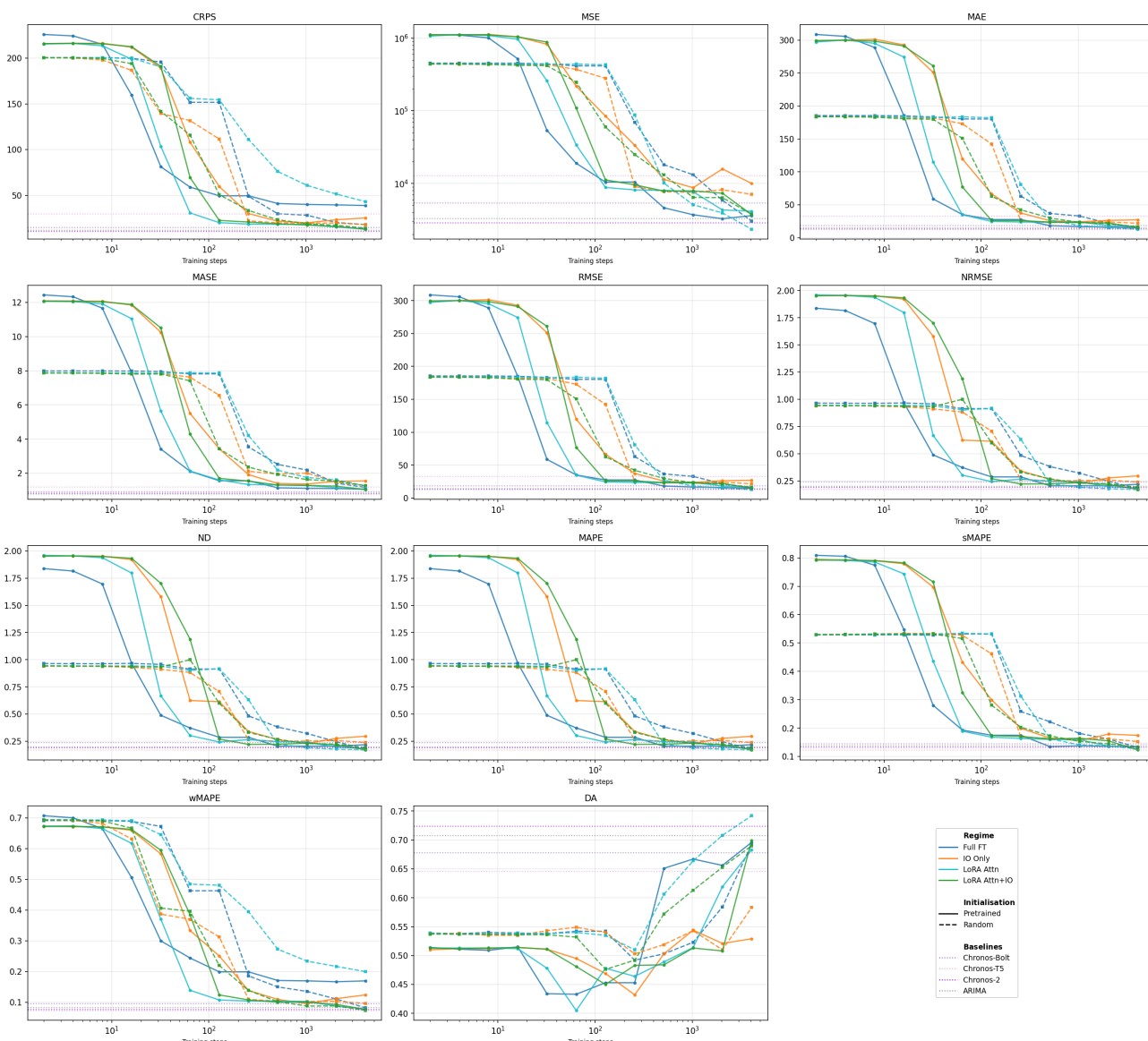

*Figure 6.* Training progression of all $h = 1$ forecasting metrics across training steps for four training regimes. Solid lines denote language-pretrained initialization (Qwen3-0.6B); dashed lines denote random initialization. Horizontal dotted lines indicate baseline performance (Chronos-T5, Chronos-Bolt, Chronos-2, ARIMA). Language-initialized models consistently converge earlier than their randomly initialized counterparts across all metrics, with the gap most pronounced in the first 100 steps. Both initializations converge in performance by 4096 steps, reaching levels competitive with established forecasting baselines.

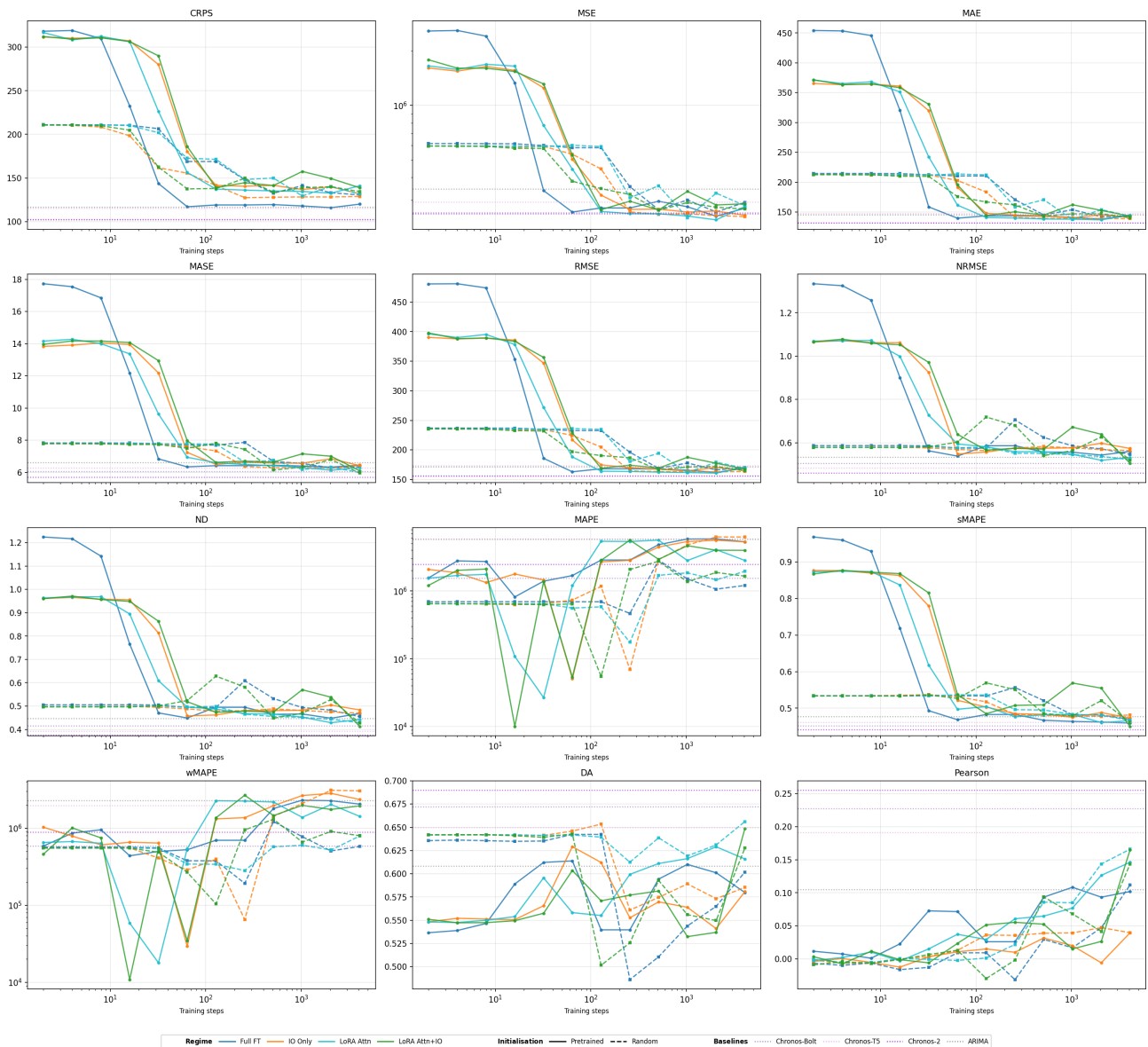

*Figure 7.* Training progression of all $h = 64$ forecasting metrics across training steps for four training regimes. Solid lines denote language-pretrained initialization (Qwen3-0.6B); dashed lines denote random initialization. Horizontal dotted lines indicate baseline performance (Chronos-T5, Chronos-Bolt, Chronos-2, ARIMA). Language-initialized models consistently converge earlier than their randomly initialized counterparts across all metrics, with the gap most pronounced in the first 100 steps. Both initializations converge in performance by 4096 steps, reaching levels competitive with established forecasting baselines.

# B. Linear Mapping Experiment

*Table 6.* Fair top-$K$ comparison across ablation conditions. For each condition, we take the best-$K$ unique matches (deduplicated) and report mean nearest-neighbor MSE. This controls for diversity: conditions with fewer unique matches are only compared at $K$ values they can support. "—" indicates fewer than $K$ unique matches available. Text + PT achieves the lowest MSE at every $K$.

| $K$ | text + PT | text + RandInit | rand + PT | rand + RandInit |
|---|---|---|---|---|
| 4 | **0.254** | 0.383 | 0.330 | 0.412 |
| 54 | **0.350** | 0.721 | — | 0.755 |
| 99 | **0.383** | 0.825 | — | 0.862 |
| 200 | **0.441** | 0.971 | — | 1.004 |
| 439 | **0.535** | — | — | — |
| 686 | **0.639** | — | — | — |

# C. Reuse vs. Reinvention: Additional Figures

**Method details.** We generate five length-512 synthetic inputs: a sine wave (period 64), a square wave (period 64), a sawtooth wave (period 64), a two-frequency signal $\sin(2\pi t/64) + 0.5\sin(2\pi t/17)$, and a linear trend with oscillation $t/T + 0.3\sin(2\pi t/80)$. Each input is independently $z$-score normalized, clipped to $[-5, 5]$, and uniformly binned into 1024 tokens using the same discretization as in the forecasting experiments. After passing the tokenized sequence through each model, we extract hidden states and fit PCA independently for each model–input pair; therefore, the plots compare within-trajectory structure and variance explained rather than absolute PCA axes across models. We exclude the first five positions to reduce attention-sink effects and color points by position modulo the dominant period of the input.

For the phase-coherence analysis, we compute Euclidean distances $d_{ij}$ between hidden states at positions $i$ and $j$ in the full 1024-dimensional hidden-state space, again excluding the first five positions. For an input with period $P$, positions are treated as same-phase when $i \bmod P = j \bmod P$. We define

$$\text{coherence} = \frac{\text{mean}(d_{ij} \mid i \bmod P = j \bmod P)}{\text{mean}(d_{ij} \mid \text{all pairs})}. \quad (4)$$

If the model encodes periodic structure, same-phase hidden states should be close relative to arbitrary pairs, so lower values indicate stronger phase structure. In the training-dynamics figure we plot $1-\text{coherence}$ so that higher values correspond to stronger periodic encoding.

# D. Cross-Domain Feature Analysis via Crosscoders

The circuit-level analysis in Appendix E shows that specific components are shared between time-series periodicity prediction and repetitive language modeling. Here we complement that *component-level* analysis with a *feature-level* analysis using crosscoders—sparse autoencoders with a shared encoder applied across domains—to discover individual latent features that fire on both time-series inputs and semantically coherent natural-language passages.

## D.1. Method

**Setup.** We train one *linear crosscoder* per transformer layer of Qwen3-0.6B. Each crosscoder has a shared encoder $\mathbf{W}_{\text{enc}} \in \mathbb{R}^{1024 \times 4096}$ followed by Top-$K$ sparsity ($K{=}64$), and two per-domain decoders $\mathbf{W}_{\text{dec}}^{(\text{PT})}, \mathbf{W}_{\text{dec}}^{(\text{FT})} \in \mathbb{R}^{4096 \times 1024}$ (12.6 M parameters total, float32). The encoder is applied independently to each domain's hidden states; because the weights are shared, the same latent feature can fire on both pretrained (PT) and finetuned (FT) inputs.

**Domains.** *PT*: Time series formatted as space-separated decimal strings (e.g., `0.123 -0.456 1.789...`), tokenized by the Qwen3-0.6B tokenizer. Sub-tokens corresponding to each timestep value are mean-pooled to produce one 1024-dim vector per timestep. *FT*: The same time series normalized per-window ($z$-score), clipped to $[-5, 5]$, and uniformly binned into 1024 tokens for the finetuned model.

**Training.** Input: 3,000 windows ($T{=}512$) from GiftEval (Aksu et al., 2024), precomputed as memory-mapped hidden-state arrays. Loss: per-domain MSE between normalized inputs and decoder reconstructions, summed over PT and FT. Dead-feature recovery (AuxK; top-64 among features firing $<1\%$ over the last 1,000 steps, weight $\frac{1}{32}$, active after step 1,000). AdamW, lr$=3\times10^{-4}$, 500 warmup steps, cosine decay over 10,000 steps. Early stopping after 1,500 steps without validation improvement (minimum step 2,000).

**Feature analysis pipeline.** After training, 50,000 time-series windows are encoded; for each of the 4,096 features we record the PT and FT firing rates (fraction of windows with non-zero activation). Features firing $\geq 1\%$ in both domains are labeled PT_FT and ranked by balance $= \min(\text{rate}_{\text{PT}}, \text{rate}_{\text{FT}})$. For the top-30 balanced features, we extract the 10 highest-activating time-series windows and 10 highest-activating WikiText-103 passages (30,000 sequences, 512 tokens each, encoded through the PT model and the shared crosscoder encoder with separate normalization statistics).

## D.2. Results

Below we present four features with the clearest cross-domain links: each fires on a specific time-series pattern *and* on thematically coherent WikiText passages.

**Feature 1712 (Layer 10) — Quantitative magnitude transitions.** PT firing rate: 50.4%, FT: 41.8%, WikiText: 14.1%. This feature detects step changes and regime shifts in time series—values that jump from a near-zero baseline to a sustained high plateau—and fires on text passages dense with numerical measurements and unit conversions (Figure 14).

Top-5 WikiText passages at the peak-activation token:

1. *(act=14.1)* "...becoming later in the year by about two days every 243-year cycle. Transits usually occur in pairs, on nearly the same date eight years apart."
2. *(act=13.8)* "In practice, forward premiums and discounts are quoted as annualized percentage deviations from the spot exchange rate..."
3. *(act=13.7)* "Wages are reflective of the type of jobs available locally, including higher than average employment in manufacturing and the public sector. The working age population of the town in 2011..."

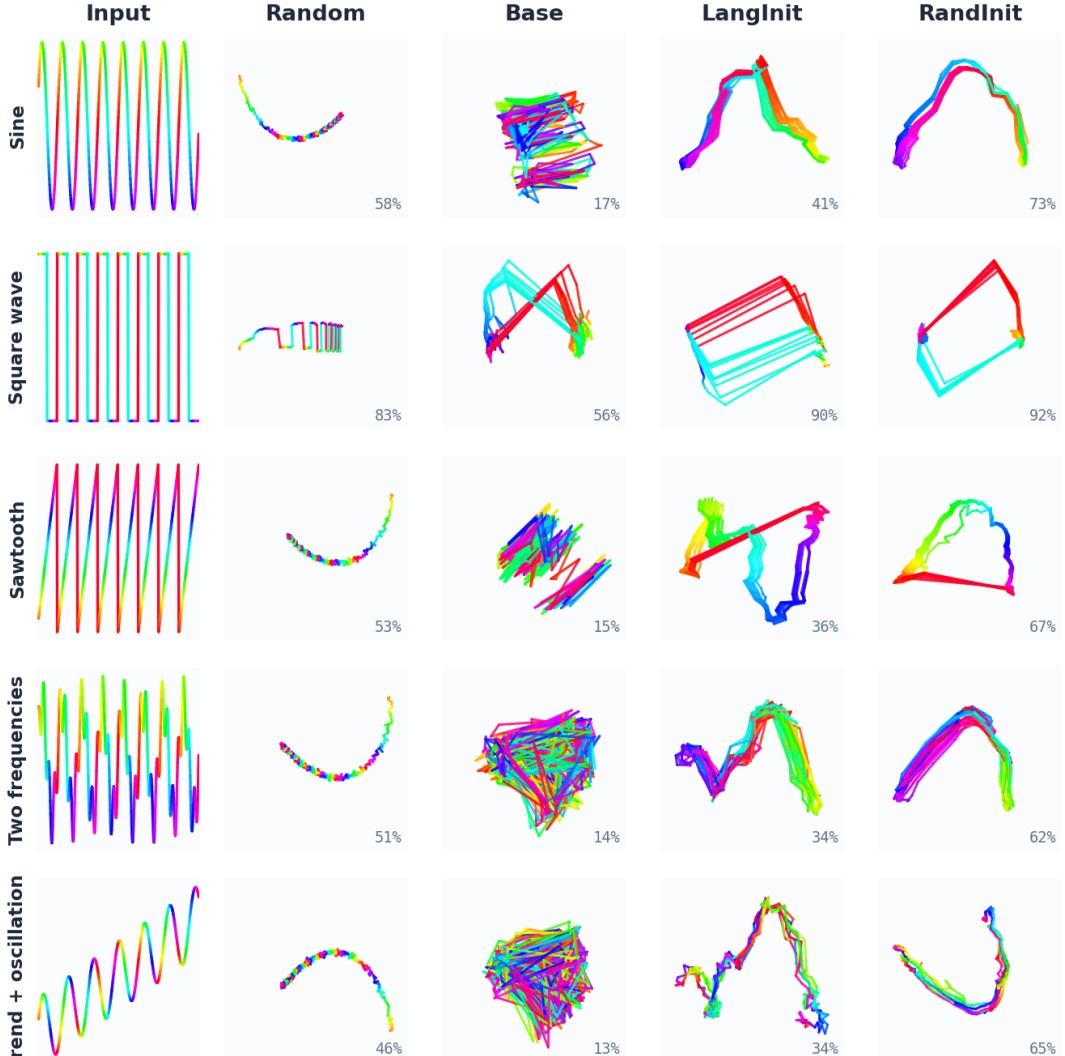

*Figure 8.* **Hidden-state trajectories for synthetic inputs at Layer 13 (2 PCA).** Trajectories are 2D PCA projections of hidden states, colored by input phase. Percentages show variance captured by 2 PCs. Random and Base produce unstructured trajectories. RandInit discovers clean, low-dimensional representations (62–92% PCA variance) while LangInit creates geometrically complex but structured trajectories (34–98%) that vary by input type.

4. *(act=13.7)* "So for americium-241, the resistivity at 4.2 K increases with time from about 2 μOhm·cm to 10 μOhm·cm after 40 hours, and saturates at about 16 μOhm·cm..."

5. *(act=13.7)* "Falcon's Fury can theoretically accommodate 800 riders per hour. Carbon-fiber wings buttress each end of a group of seats..."

The shared representation encodes *quantitative magnitude and transition*: literal level shifts in time series, and passages dense with measurements, unit conversions, and numerical comparisons in text.

**Feature 2469 (Layer 9) — Tropical weather systems.**
PT: 29.9%, FT: 20.7%, WikiText: 12.6%. Fires on volatile, regime-switching time series and exclusively on tropical cyclone and hurricane narratives in text (Figure 15).

Top-5 WikiText passages:

1. *(act=8.7)* "...the mean locus of formation shifts westward to the Caribbean and Gulf of Mexico, reversing the eastward progression of June through August. Wind shear from westerlies increases substantially through November..."

2. *(act=8.3)* "...due to a combination of very high wind shear and dry air. By October 17, most of the deep convection associated with the system dissipated; however, a brief decrease in wind shear allowed Omar to re-strengthen..."

3. *(act=8.1)* "The wave continued westward and related thunderstorm activity increased during the following week. The convective system organized into Tropical Depression Twenty-E on September 28..."

4. *(act=8.1)* "A tropical wave moved across the northeast Pacific Ocean and formed a tropical depression south of Mexico on October 16. It strengthened at a moderate pace and reached

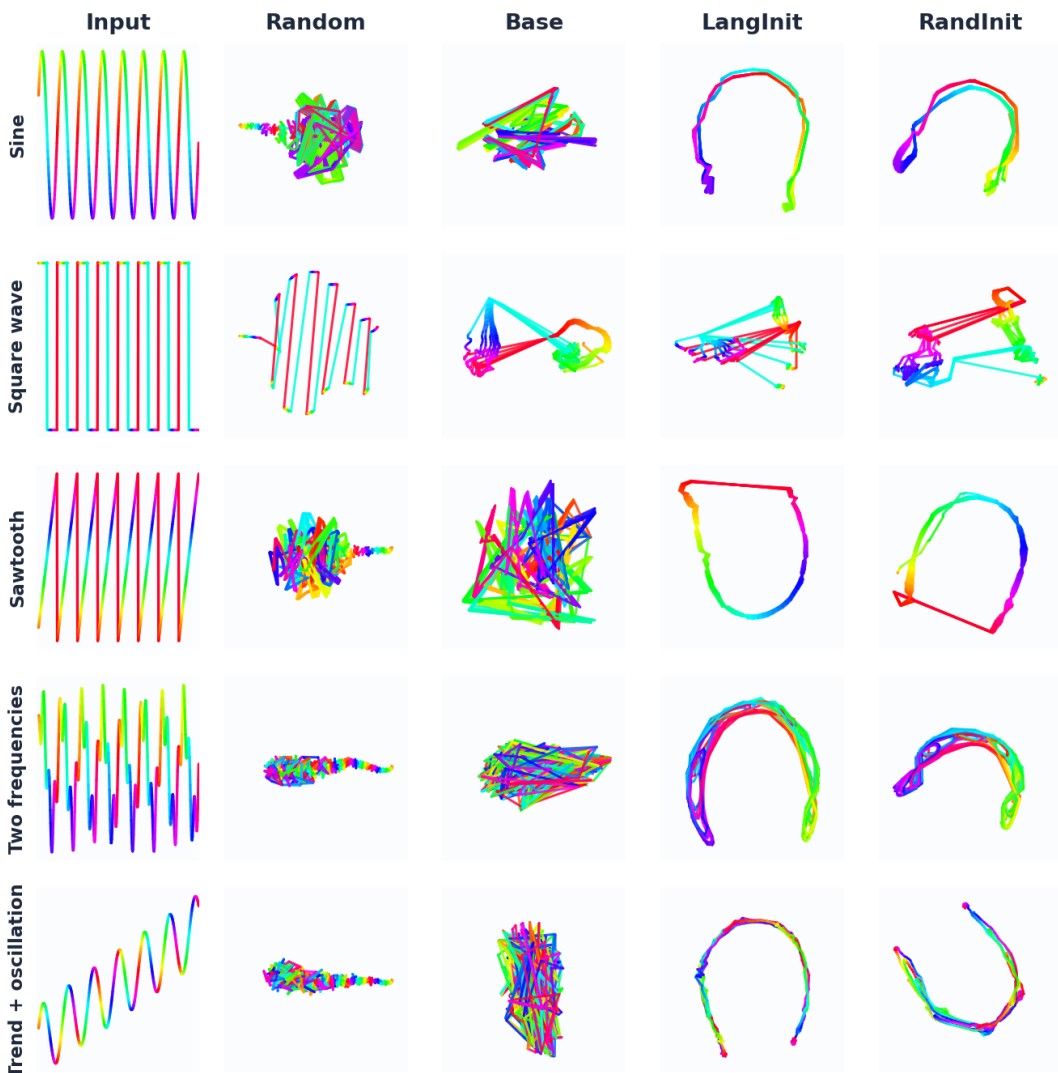

*Figure 9.* **Hidden-state trajectories for synthetic inputs at Layer 13 (t-SNE).** Same setup as Figure 8 but using t-SNE (perplexity 30) instead of PCA. Despite the different global geometries revealed by PCA, t-SNE shows that LangInit and RandInit develop similar local neighborhood structure: periodic inputs form smooth, phase-ordered curves in both models, confirming that both arrive at functionally equivalent representations through different geometric paths.

hurricane intensity on October 18."

5. *(act=7.9)* "...formation of Typhoon Chanchu in the western Pacific enhanced convective activity over the Bay of Bengal. By April 22, a trough developed along an axis from the southern Bay of Bengal eastward to the Andaman Sea."

The time-series patterns—volatile signals with sudden regime changes—mirror the physical phenomena described in the text: tropical storms that intensify, weaken under wind shear, and shift track.

**Feature 3888 (Layer 7) — Naval battle events.** PT: 18.5%, FT: 26.3%, WikiText: 10.5%. Fires on sharp isolated spikes in otherwise stable time series and on naval/military battle narratives with precise timestamps (Figure 16).

Top-5 WikiText passages:

1. *(act=10.1)* "King George V had only 32 percent of her fuel left while Rodney had only enough fuel to continue the chase at high speed until 8:00 the following day. Admiral Tovey signalled his battlegroup..."

2. *(act=9.3)* "At 7:20 on 19 July, the destroyer force spotted and was spotted by a pair of Italian light cruisers; Giovanni dalle Bande Nere and Bartolomeo Colleoni, which opened fire seven minutes later."

3. *(act=9.2)* "Shortly before 16:00 the battlecruisers of I Scouting Group encountered the British 1st Battlecruiser Squadron under the command of Vice Admiral David Beatty. The opposing ships began an artillery..."

4. *(act=9.2)* "The eastern wind was not communicated to the aircraft, but was 270°, varying from 20 to 40 knots (37 to 74 km/h). The take-off started at 14:42:43..."

5. *(act=9.1)* "...torpedo boat attacks and at 07:30, Burrough sent

**FT — PCA of sine wave (period=64)**

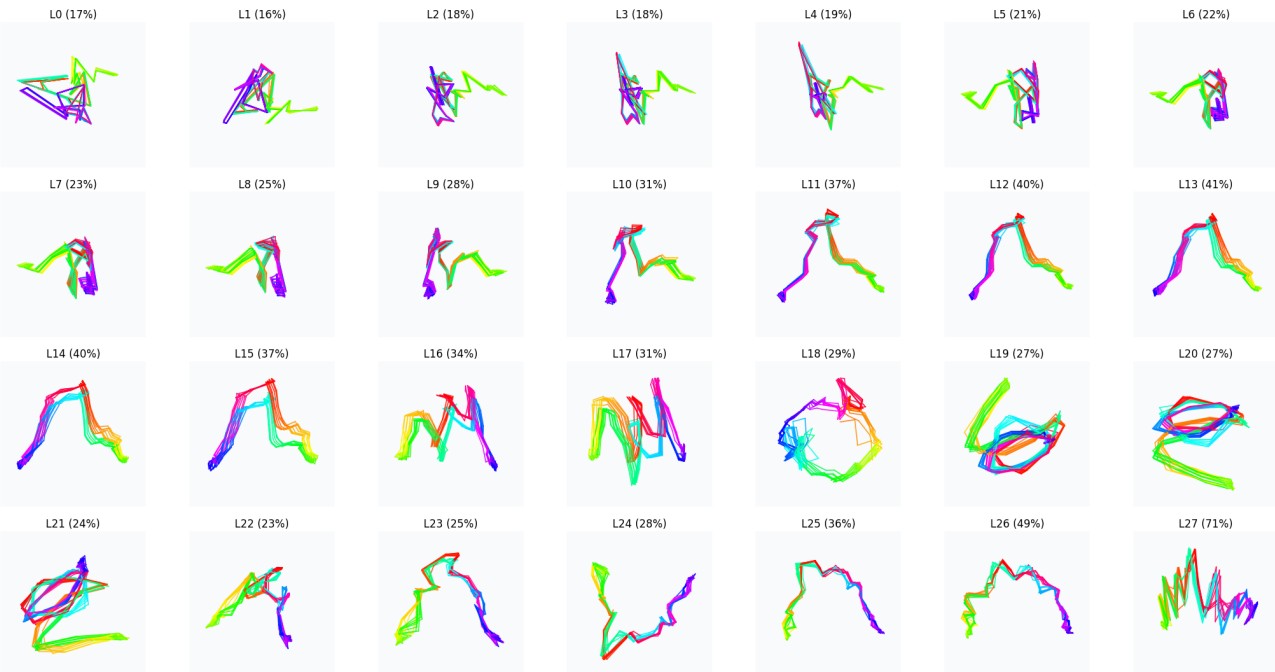

*Figure 10.* **LangInit: sine wave PCA trajectories across all 28 layers.** Each subplot shows the 2D PCA projection of hidden states from a sine wave (period 64) at one transformer layer, colored by input phase. Percentages show variance explained by 2 PCs. LangInit exhibits layer-specific geometry with varying complexity and moderate PCA variance (17–49%), reflecting the rich, heterogeneous representations inherited from language pretraining.

Eskimo and Somali back to help Manchester but they arrived too late, took on survivors..."

Both modalities encode *sudden, precisely-located events*: an anomalous spike at a single timestep in time series, and a precisely-timestamped combat event in text.

**Feature 2567 (Layer 8) — Missing / null data.** PT: 35.9%, FT: 36.8%, WikiText: 9.4%. This feature fires exclusively on NaN/missing time-series data (all 10 top-activating windows are entirely NaN, with peak activation 40.3) and on `<unk>` tokens and incomplete references in text.

Top-5 WikiText passages:

1. *(act=19.8)* "...at the Royal Navy School of Flight Deck Operations at RNAS Culdrose. The following is an incomplete list of some of the surviving aircraft."
2. *(act=18.5)* "`<unk>`, `<unk>`, `<unk>`, `<unk>`, `<unk>`, Ulaid. Slightly later major groups included the Connachta, `<unk>`, `<unk>`. Smaller groups included the `<unk>`..."
3. *(act=18.0)* "...he encountered bad weather, forcing him to return to Japan with heavy damage. Without waiting for Vizcaino, another ship—built in Izu by the Tokugawa shogunate..."
4. *(act=17.8)* "Luke 9: `<unk>`-`<unk>` — καὶ `<unk>`, `<unk>` `<unk>` `<unk>` πνεύματος `<unk>` `<unk>`..."
5. *(act=17.7)* "...whom he married in the late 250s when she was 17 or 18 years old. The number of children Odaenathus had with his first wife is unknown and only one is attested."

The model represents *absent information* identically across modal-ities: NaN values in time series and `<unk>` tokens in text both occupy the same region of representation space.

## E. Causal Circuit Identification

The correlational analyses in Section **??** show that finetuning reuses pretrained directions and that cross-domain features exist. Here we present a preliminary causal analysis: we identify specific model components responsible for periodic time-series prediction and test what role those components play in language modeling.

### E.1. Method

Since IO-only finetuning only trains embedding and LM-head layers, its 28 transformer layers are identical to PT. We zero-ablate ([Wang et al., 2024](#)) each of the 476 components (448 attention heads + 28 MLPs) on IO-only finetuning and measure the loss increase ($\Delta\mathcal{L}$) on periodic time series versus non-periodic controls.

**Zero-ablation implementation.** For each attention head, we register a `forward_pre_hook` on the output projection (`o_proj`) that zeros the head's 128-dimensional slice of the concatenated 2048-dimensional head output before projection. For each MLP layer, we register a `forward_hook` that replaces the MLP output with zeros, so the residual stream passes through unchanged. All ablations are performed under `torch.no_grad()`.

**RI — PCA of sine wave (period=64)**

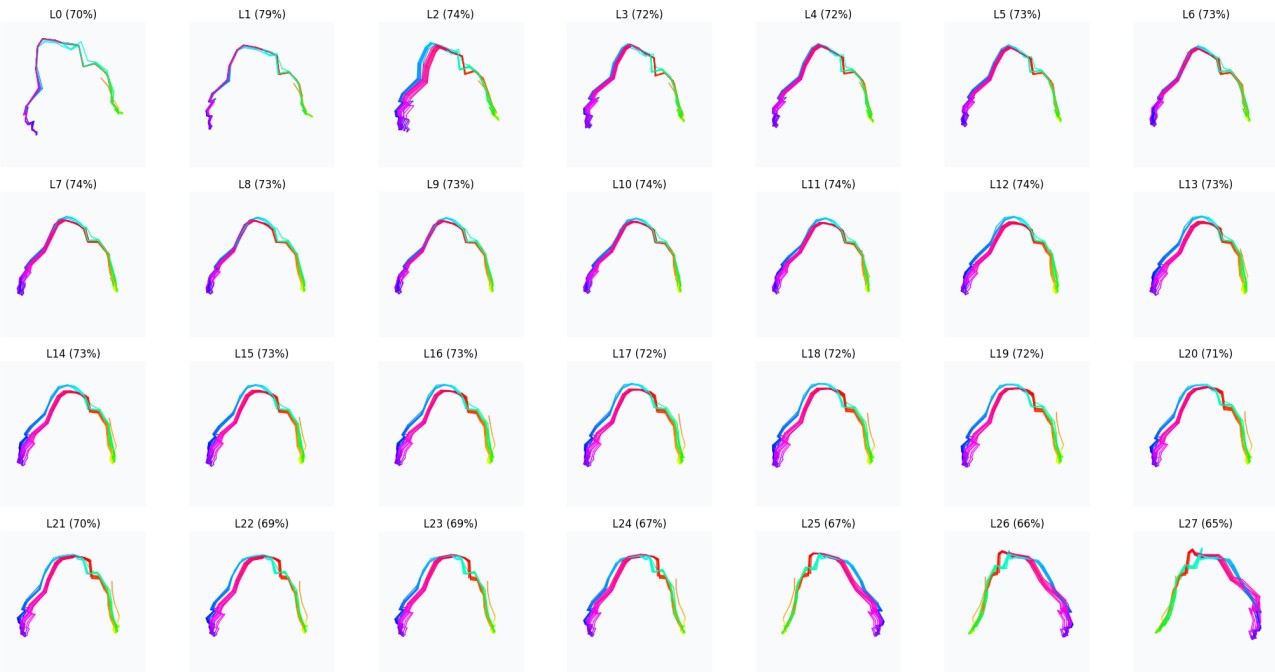

*Figure 11.* **RandInit: sine wave PCA trajectories across all 28 layers.** Same setup as Figure 10. RandInit produces nearly identical clean arcs at every layer with uniformly high PCA variance (67–79%), confirming that its learned representation is geometrically homogeneous across the network.

**Synthetic time series.** We generate 20 windows (length 512) for each of nine synthetic types. *Periodic*: sine, square wave, sawtooth, seasonal (trend + oscillation), damped sine. *Control*: white noise, constant, linear trend, random walk. Each window is independently $z$-score normalized, clipped to $[-5, 5]$, and uniformly binned into 1024 tokens (matching the IO-only finetuning tokenizer). This yields 180 evaluation windows total (100 periodic, 80 control).

**Selectivity metric.** For each ablated component we compute:

$$\Delta\mathcal{L}_{\text{periodic}} = \bar{\mathcal{L}}_{\text{ablated}}^{\text{periodic}} - \bar{\mathcal{L}}_{\text{baseline}}^{\text{periodic}}, \tag{5}$$

$$\Delta\mathcal{L}_{\text{control}} = \bar{\mathcal{L}}_{\text{ablated}}^{\text{control}} - \bar{\mathcal{L}}_{\text{baseline}}^{\text{control}}, \tag{6}$$

$$\text{selectivity} = \Delta\mathcal{L}_{\text{periodic}} - \Delta\mathcal{L}_{\text{control}}. \tag{7}$$

Components with high $\Delta\mathcal{L}_{\text{periodic}}$ and high selectivity are specifically critical for periodic prediction rather than general-purpose.

**Cumulative ablation.** We compose multiple ablation hooks simultaneously using nested context managers. For the pairwise sweep, we test all $\binom{8}{2} = 28$ pairs of the top-8 selective components. For the growing cumulative, we add components one at a time in order of individual $\Delta\mathcal{L}_{\text{periodic}}$. The superadditivity ratio is $\rho = \Delta\mathcal{L}_{\text{combined}} / \sum_i \Delta\mathcal{L}_i^{\text{individual}}$ (Conmy et al., 2023; Elhage et al., 2021); we use a threshold of $\rho > 1.2$ to account for noise.

**WikiText transfer.** We ablate the top-5 circuit components simultaneously on 2,000 WikiText-103 passages (length 512) through the pretrained model (which shares identical transformer

layers with IO-only finetuning). For each passage we record the per-sequence cross-entropy loss before and after ablation. The 50 most-degraded and 50 least-degraded passages are extracted for qualitative analysis.

### E.2. Results

**Individual component sweep.** Table 7 shows the top-8 components ranked by $\Delta\mathcal{L}_{\text{periodic}}$. The two most impactful are both in Layer 1: $\text{MLP}_{\text{L1}}$ ($\Delta\mathcal{L}_p = 5.75$, selectivity $= 3.44$) and head L1H4 ($\Delta\mathcal{L}_p = 4.50$, selectivity $= 3.81$). Head L20H0 is the third most impactful with the highest selectivity (3.28).

**Cumulative ablation reveals a composed circuit.** The Layer 1 pair (head L1H4 + $\text{MLP}_{\text{L1}}$) is strongly superadditive: ablating them together produces $\Delta\mathcal{L}_p = 15.50$, which is 51% larger than the sum of their individual effects ($\rho = 1.51$; Table 8). No other pair among the 28 tested exceeds the $\rho > 1.2$ threshold. Adding head L20H0 maintains superadditivity ($\rho = 1.36$); beyond three components, returns become subadditive ($\rho < 1$), indicating compensation rather than composition.

**The periodicity circuit in language.** We ablate the top-5 circuit components simultaneously on 2,000 WikiText-103 passages through the pretrained model. The circuit ablation causes a mean loss increase of $\Delta\mathcal{L} = 4.53$ nats—two orders of magnitude larger than any individual head's WikiText effect ($\leq 0.04$), confirming circuit-level interaction.

The most-degraded passages ($\Delta\mathcal{L} = 8$–9; Table 9) are dominated

FT — Sine wave PCA across training

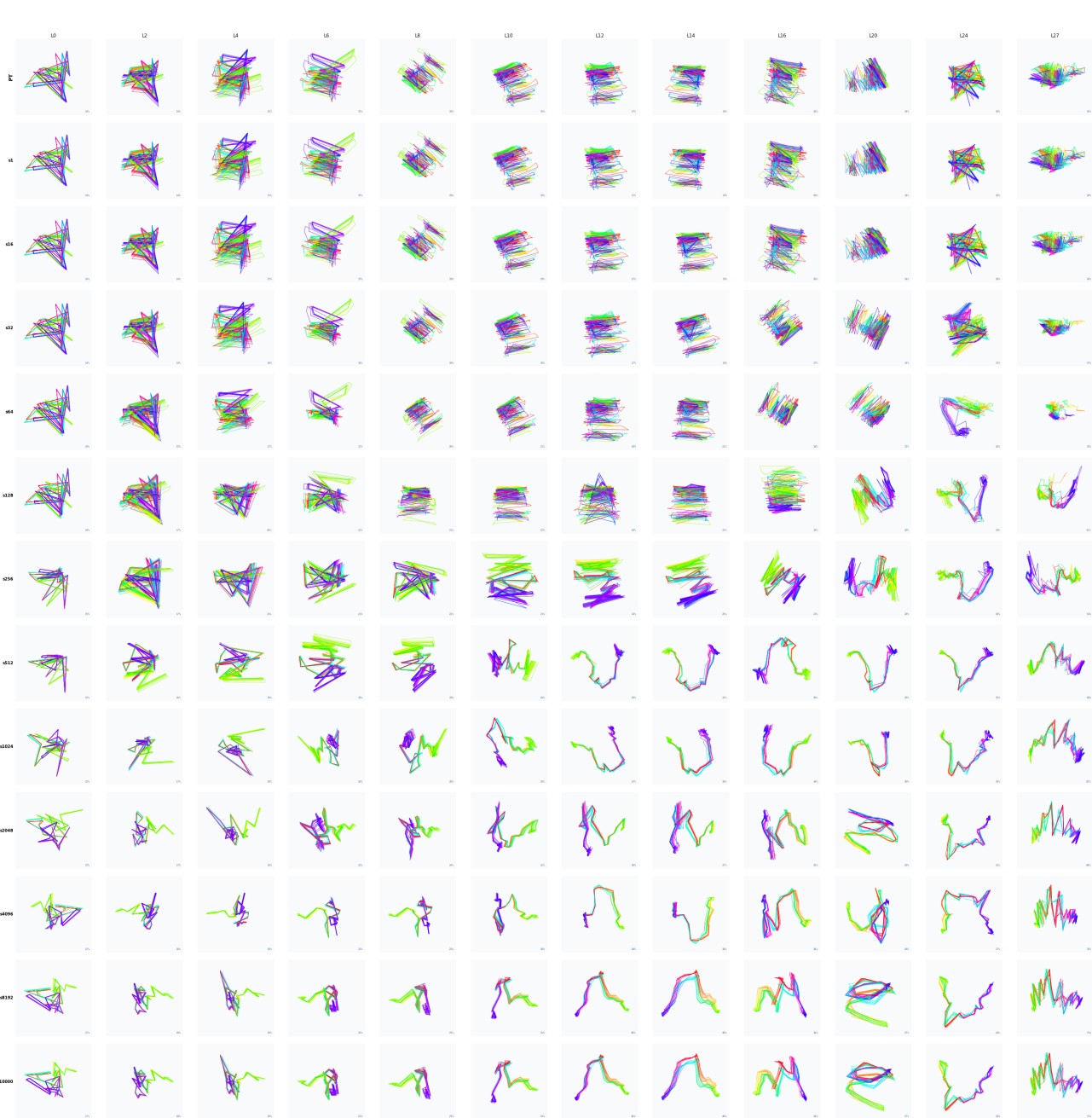

*Figure 12.* **LangInit: sine wave representations across training checkpoints.** Rows are training steps (top: PT baseline; bottom: final checkpoint at step 10,000), columns are selected layers. LangInit begins from the pretrained model's complex trajectories and gradually reshapes them into structured loops, with the transition occurring around steps 256–1024.

by sequential enumerations and repeating grammatical templates: biographical sequences ("married to...she gave birth to..."), game mechanics with parallel clause structure ("sets a task for each stage...this task must be completed...the player with the most"), Billboard chart progressions, award category listings, and structured Wikipedia sections with repeating headers. In contrast, the least-degraded passages ($\Delta\mathcal{L} < 2$) are dense, non-repetitive prose: ecclesiastical titles, academic citations, canonical law, and liter-

ary criticism—text where predicting the next token depends on semantic content rather than structural repetition.

This suggests the circuit tracks *sequential repetitive structure*—the same abstract property shared by periodic time series and repetitive text. However, the evidence is preliminary: the WikiText passages most degraded by the circuit do not all show an obvious repetitive pattern, and disentangling the circuit's role in general sequence modeling from periodicity-specific computation requires further

RI — Sine wave PCA across training

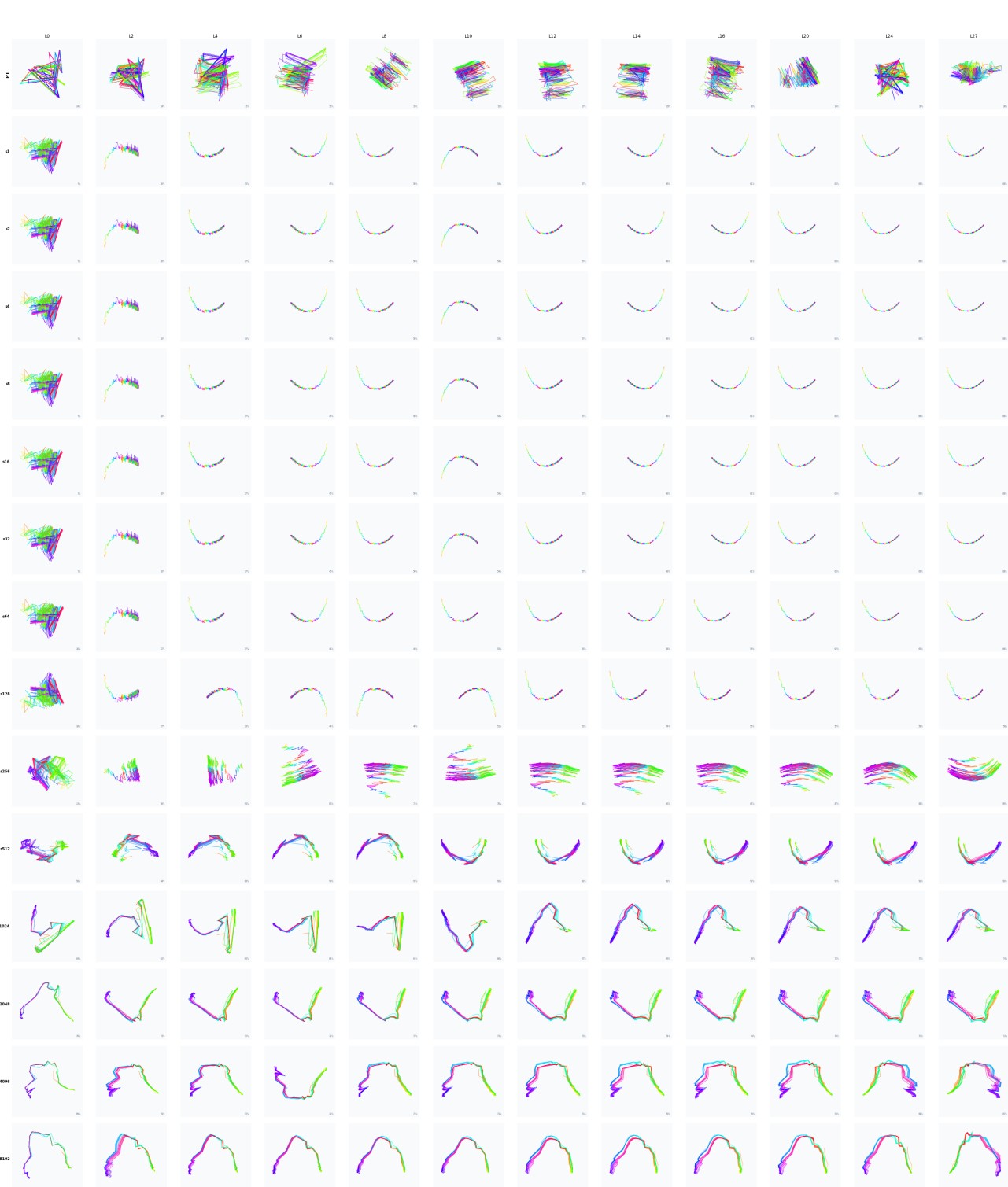

*Figure 13.* **RandInit: sine wave representations across training checkpoints.** Same setup as Figure 12. RandInit starts from near-empty representations (random weights) and converges to uniform clean arcs by step 2048, with all layers collapsing to the same geometry simultaneously.

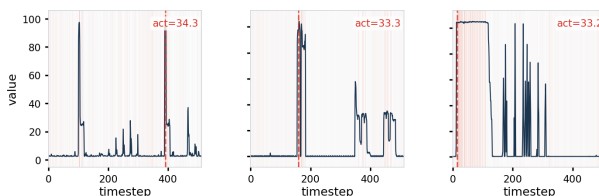

*Figure 14.* **Feature 1712 (Layer 10): Quantitative magnitude transitions.** Top-3 activating time-series windows. Blue: raw signal; red dashed: peak-activation timestep; orange shading: activation intensity. All three windows share a sudden jump from a low baseline to a high-magnitude plateau.

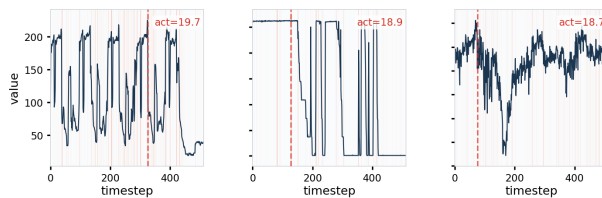

*Figure 15.* **Feature 2469 (Layer 9): Tropical weather systems.** Top-3 activating windows. The left window shows high-variance oscillations with abrupt drops; the middle and right show diverse volatile patterns with regime switching.

investigation.

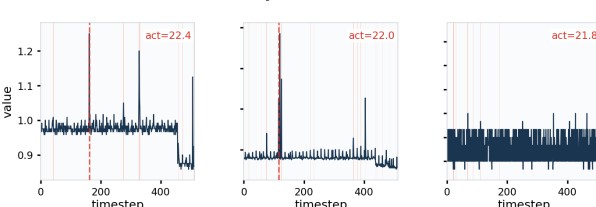

*Figure 16.* **Feature 3888 (Layer 7): Naval battle events.** Top-3 activating windows. Each shows a low-variance baseline punctuated by sharp spikes at the peak-activation timestep (red dashed line).

*Table 3.* Complete hyperparameter configuration for all experiments. All runs share the same configuration except where noted.

| Category | Parameter | Value |
|---|---|---|
| Model | Base model | Qwen3-0.6B |
| | Architecture | Qwen3ForCausalLM and Qwen defaults |
| Tokenizer | Scaling | Z-score (mean/std from context) |
| | Binning | Uniform |
| | Vocab size ($V$) | 1024 |
| | Bin range | $[-5, 5]$ |
| | Use EOS token | No |
| Training | Optimizer | AdamW |
| | Learning rate | $1 \times 10^{-4}$, $3 \times 10^{-4}$, $1 \times 10^{-3}$, or $3 \times 10^{-3}$ |
| | LR schedule | Linear warmup + cosine decay |
| | Warmup ratio | 3% |
| | LR end factor | $1 \times 10^{-4}$ |
| | Precision | bf16 (mixed) |
| Batching | Effective batch size | 128 |
| | Epoch length | 5,000 steps |
| Data | Dataset | GiftEval Pretrain (152 sub-datasets) |
| | Context length | 512 |
| | Target length | 64 |
| | Window stride | 600 |
| Loss | Loss function | Quantile (pinball) loss |
| | Quantiles | $\{0.1, 0.2, \ldots, 0.9\}$ |
| | Softmax temperature | $10^{-2}$ |
| Evaluation | Eval samples | 1,000 extracted from the dataset (see script) |
| | Eval horizons | $h \in \{1, 64\}$ |
| | Eval strategy | Autoregressive |
| LoRA | Rank ($r$) | 4, 8 |
| | Alpha ($\alpha$) | 16, 32 |
| | Dropout | 0.05 |
| | Bias | None |
| | Targets (Attn) | `q_proj, k_proj, v_proj, o_proj` |
| Reproducibility | Random seed | 420 |
| | Seeded modules | Python, NumPy, PyTorch (CPU + CUDA) |
| | Compute Usage | About 12 hours on 8 Nvidia A100 per model |

*Table 4.* Single-step ($h=1$) forecasting performance on the held-out evaluation set. All NanoTS variants use Qwen3-0.6B at training step 128. Best value per metric within each section is **bolded**. ↑: higher is better; ↓: lower is better.

| | Model | CRPS↓ | MSE↓ | MAE↓ | MASE↓ | RMSE↓ | NRMSE↓ | ND↓ | MAPE↓ | sMAPE↓ | wMAPE↓ | DA↑ |
|---|---|---|---|---|---|---|---|---|---|---|---|---|
| Pretrained | Full Finetune | 49.51 | 10461 | 27.60 | **1.561** | 27.60 | 0.286 | 0.286 | 0.286 | 0.175 | 0.199 | 0.453 |
| | IO Only | 59.56 | 84846 | 66.53 | 3.430 | 66.53 | 0.614 | 0.614 | 0.614 | 0.300 | 0.250 | 0.469 |
| | LoRA Attn | **20.10** | **8818** | **24.68** | 1.607 | **24.68** | **0.243** | **0.243** | **0.243** | **0.168** | **0.107** | **0.478** |
| | LoRA Attn+IO | 22.54 | 11220 | 26.11 | 1.702 | 26.11 | 0.270 | 0.270 | 0.270 | 0.174 | 0.124 | 0.450 |
| Random Init | Full Finetune | 151.8 | 415339 | 180.4 | 7.819 | 180.4 | 0.914 | 0.914 | 0.914 | 0.532 | 0.463 | **0.542** |
| | IO Only | 111.6 | 280932 | 142.3 | 6.565 | 142.3 | 0.706 | 0.706 | 0.706 | 0.462 | 0.313 | 0.540 |
| | LoRA Attn | 154.5 | 430387 | 182.2 | 7.872 | 182.2 | 0.917 | 0.917 | 0.917 | 0.532 | 0.481 | 0.535 |
| | LoRA Attn+IO | 50.68 | **59864** | **62.90** | **3.438** | **62.90** | **0.599** | **0.599** | **0.599** | **0.281** | **0.220** | 0.476 |
| Baselines | Chronos-Bolt | 14.94 | 5409 | 18.91 | 0.934 | 18.91 | 0.240 | 0.240 | 0.240 | 0.146 | 0.096 | 0.678 |
| | Chronos-T5 | 29.88 | 12764 | 32.99 | 1.212 | 32.99 | **0.173** | **0.173** | **0.173** | **0.124** | **0.073** | 0.646 |
| | Chronos-2 | **10.89** | **2886** | **13.38** | 0.830 | **13.38** | 0.192 | 0.192 | 0.192 | 0.133 | 0.077 | **0.724** |
| | ARIMA | 11.94 | 3327 | 14.86 | **0.805** | 14.86 | 0.201 | 0.201 | 0.201 | 0.138 | 0.083 | 0.708 |
| | Qwen3 Text | 213.5 | 2.6M | 213.5 | 376667 | 213.5 | 39508 | 39508 | 39508 | 0.563 | 19754 | 0.503 |

*Table 5.* Multi-step ($h=64$) forecasting performance on the held-out evaluation set. All NanoTS variants use Qwen3-0.6B at training step 128. Best value per metric within each section is **bolded**. ↑: higher is better; ↓: lower is better.

| | Model | CRPS↓ | MSE↓ | MAE↓ | MASE↓ | RMSE↓ | NRMSE↓ | ND↓ | MAPE↓ | sMAPE↓ | wMAPE↓ | DA↑ | Pearson↑ |
|---|---|---|---|---|---|---|---|---|---|---|---|---|---|
| Pretrained | Full Finetune | **119.0** | 271341 | 143.7 | **6.424** | 168.2 | 0.586 | 0.495 | **2.85M** | **0.483** | 696208 | 0.540 | 0.026 |
| | IO Only | 141.5 | 319226 | 147.9 | 6.502 | 174.9 | **0.557** | 0.462 | 2.68M | 0.503 | 1.32M | **0.612** | 0.015 |
| | LoRA Attn | 137.2 | **259652** | 140.8 | 6.602 | **164.0** | 0.581 | 0.487 | 5.39M | 0.505 | 2.27M | 0.555 | 0.029 |
| | LoRA Attn+IO | 138.8 | 265546 | 144.0 | 6.628 | 169.0 | 0.566 | 0.473 | 2.82M | 0.485 | 1.36M | 0.571 | **0.052** |
| Random Init | Full Finetune | 168.9 | 583696 | 210.5 | 7.700 | 232.7 | 0.575 | 0.494 | 693452 | 0.534 | 377358 | 0.642 | 0.009 |
| | IO Only | 141.3 | 445540 | 183.5 | **7.341** | 205.0 | **0.563** | **0.478** | 1.17M | **0.517** | 394850 | **0.653** | **0.036** |
| | LoRA Attn | 171.4 | 592435 | 212.3 | 7.770 | 234.7 | 0.579 | 0.499 | 583786 | 0.536 | 339884 | 0.639 | 0.001 |
| | LoRA Attn+IO | **137.9** | **346961** | **166.8** | 7.812 | **190.5** | 0.719 | 0.628 | **55012** | 0.569 | **104026** | 0.502 | −0.030 |
| Baselines | Chronos-Bolt | **101.9** | **251450** | **131.7** | 6.068 | 156.8 | 0.504 | 0.416 | **1.55M** | 0.452 | **587680** | 0.672 | 0.227 |
| | Chronos-T5 | 115.2 | 292023 | 147.4 | 6.291 | 173.7 | 0.484 | 0.393 | 5.84M | 0.462 | 1.98M | 0.650 | 0.191 |
| | Chronos-2 | 102.4 | 255127 | 132.0 | **5.709** | **156.2** | **0.460** | **0.376** | 2.48M | **0.441** | 888025 | **0.690** | **0.255** |
| | ARIMA | 116.6 | 345041 | 145.4 | 6.609 | 171.8 | 0.534 | 0.446 | 5.80M | 0.478 | 2.27M | 0.608 | 0.105 |
| | Qwen3 Text | 489.1 | 213.1M | 489.1 | 9911.0 | 720.8 | 5672.0 | 824.6 | 3.23M | 0.568 | 1.62M | 0.589 | −0.048 |

*Table 7.* Top-8 components by $\Delta\mathcal{L}$ on periodic TS under zero-ablation. Selectivity $= \Delta\mathcal{L}_{\text{periodic}} - \Delta\mathcal{L}_{\text{control}}$ isolates periodicity-specific impact. Full sweep: 476 components.

| Component | $\Delta\mathcal{L}_{\text{periodic}}$ | $\Delta\mathcal{L}_{\text{control}}$ | Selectivity |
|---|---|---|---|
| $\text{MLP}_{\text{L1}}$ | 5.75 | 2.31 | 3.44 |
| Head L1H4 | 4.50 | 0.69 | 3.81 |
| Head L20H0 | 3.40 | 0.13 | 3.28 |
| Head L13H6 | 3.35 | 2.56 | 0.79 |
| Head L23H6 | 2.00 | 0.06 | 1.94 |
| Head L21H8 | 1.85 | −0.25 | 2.10 |
| Head L15H3 | 1.55 | 0.38 | 1.18 |
| Head L11H8 | 1.25 | 0.56 | 0.69 |

*Table 8.* Cumulative ablation. The Layer 1 pair is strongly super-additive ($\rho = 1.51$); the core circuit saturates at 3 components.

| Components | $\Delta\mathcal{L}_{\text{periodic}}$ | $\sum \Delta\mathcal{L}_{\text{indiv.}}$ | $\rho$ | $\Delta\mathcal{L}_{\text{control}}$ |
|---|---|---|---|---|
| Head L1H4 + $\text{MLP}_{\text{L1}}$ | 15.50 | 10.25 | **1.51** | 2.38 |
| + Head L20H0 (top-3) | 18.55 | 13.65 | **1.36** | 3.69 |
| + Head L21H8 (top-4) | 18.40 | 15.50 | 1.19 | 3.06 |
| + Head L23H6 (top-5) | 18.55 | 17.50 | 1.06 | 3.00 |
| All top-8 | 18.30 | 23.65 | 0.77 | 3.13 |

*Table 9.* WikiText-103 passages most and least degraded by ablation of the periodicity circuit (top-5 components). Mean $\Delta\mathcal{L} = 4.53$ across 2,000 passages.

| $\Delta\mathcal{L}$ | Passage excerpt |
|---|---|
| *Top 20 most degraded* | |
| 9.19 | "…Townsend has been married to Tracy Turner, his girlfriend since he was 19. She gave birth to thei[r]…" |
| 8.88 | "…Preston winger Will Hayhurst, a Republic of Ireland under-21 international, was signed on a one-month loan…" |
| 8.88 | "…with the NHK Symphony Orchestra, but cancelled both deals upon Mwanga's return from Japan. Mwanga immediately quit…" |
| 8.88 | "…his/her final score on the song, with money being awarded in Guitar Hero World Tour. The games have also added…" |
| 8.75 | "…Viscount, sets a task for each stage. This task must be completed before the player can continue to another map…" |
| 8.50 | "…The player character is Michel Ardan, an eccentric and intrepid French scientist who is enthusiastic, daring…" |
| 8.38 | "…Best Music Video, Long Form. In 1998, the categories were retitled Best Short Form Music Video, and Best Long…" |
| 8.31 | "…on the Billboard Hot 100 twenty-nine, the highest U.S. entry among all singles released from the album…" |
| 8.31 | "…long run, with The A.V. Club attributing much of the show's early success to the character…" |
| 8.25 | "…Yankovic recorded 'Here's Johnny', a parody of 'Who's Johnny' by El DeBarge. The song, a loving ode to…" |
| 8.19 | "…The music video of 'Crazy in Love', released in May 2003, was directed by Jake Nava and filmed…" |
| 8.19 | "…Finishing with the worst record in the NHL, Columbus had the best chance of receiving the first overall pick…" |
| 8.13 | "…while one in the Pyramid Texts says the name is based on words shouted by Osiris…" |
| 8.06 | "…the album Daydream most resembles in its emphasis on R&B grooves. Tucker specifically complimented 'One Sweet Day'…" |
| 8.00 | "…similar to Konami's Guitar Freaks and to a lesser extent Harmonix's previous music games such as Frequency…" |
| 8.00 | "…he 'hit the wall with play-along music games', and challenged the game makers to explore other ways…" |
| 7.94 | "…saying that he 'was upset. But when you see the talent that was there, it was an honour just to be in the final'…" |
| 7.94 | "…Flags indicate national team as defined under FIFA eligibility rules. Players may hold more than one non-FIFA…" |
| 7.94 | "…at the Television Critics Association summer media tour in Beverly Hills, California…" |
| 7.94 | "…co-wrote and produced a song with Kenneth 'Babyface' Edmonds, with whom she had collaborated on Music Box…" |
| *Top 10 least degraded* | |
| 0.83 | "…Porto e Santa Rufina; Sub-dean of the Sacred College of Cardinals; prefect of the S.C. of the Good Government…" |
| 1.48 | "…Cardinal-Priest of SS. Giovanni e Paolo; Grand penitentiary; prefect of the Congregation for the correction…" |
| 1.59 | "…From the Jewish Question to the Jewish State: An Essay on the Theory of Zionism (thesis), Princeton Univ.…" |
| 1.77 | "…called Saprang's transfer a 'demotion' and a 'punishment.' However, Saprang himself claimed that he did not…" |
| 1.82 | "…the Society of Jesus, provided he observed the canon law; and that it was desirable that the pope should…" |
| 1.89 | "…the abnormal excess of white blood cells in people with the clinical syndrome described by Velpeau and Bennett…" |
| 1.89 | "…The protagonist sounds like a 'colonial administrator', and his reference to seeking a newer world echoes…" |
| 1.94 | "…Mbaruk's nephew, Mbaruk bin Rashid, refused to acknowledge the appointment of a new leader…" |
| 1.98 | "…00 works were published underground over the course of the war. Literary discussions were held…" |
| 2.02 | "…Although the poem was defended by a few critics, E.C. Pettet returned to the argument that the poem lacked…" |

