# OpenReview forum: "Language Pretraining Gives Structured Forecasters a Sequential Prior"
_ICML.cc/2026/Workshop/FMSD — FMSD @ ICML 2026 Poster_

### Official Review · Reviewer_Xj1X · 2026-05-20

**Rating:** 7
**Confidence:** 3

**Review:**

- Summary: This paper studies whether language pretraining contributes something useful to time series forecasting beyond the transformer architecture itself. It adapts Qwen3 to probabilistic univariate forecasting on GiftEval using discretized continuous values and quantile loss training. The comparison is between language pretrained Qwen 3 and an identical randomly initialized Qwen 3 architecture, under full finetuning, IO only tuning, LoRA attention tuning, and LoRA IO tuning. The main finding is that language pretraining gives a large early optimization advantage, especially when adaptation is limited.

- Strengths: The paper asks a timely and important question for structured-data foundation models: when does reusing a language backbone help structured forecasting, and what is being transferred? This matches the FMSD theme very well, especially the workshop’s interest in cross-domain pretraining, time-series foundation models, and multimodal structured learning. The main experimental comparison is clean. Comparing a pretrained Qwen3 model against an identical random-initialized architecture is a good way to isolate the contribution of language pretraining from architecture and tokenization. The result that LoRA-attention recovers much of full finetuning’s transfer benefit is practically meaningful, because parameter-efficient adaptation is significant.

- Weaknesses and Questions: The paper uses one backbone, one tokenization scheme, and univariate forecasting; the authors acknowledge this limitation themselves, for a workshop paper, I think this is acceptable, but it weakens broad claims about “language pretraining gives structured forecasters a sequential prior.” The main performance gain appears to be mostly an optimization-speed gain, not necessarily a final-performance gain. The appendix states that both initialization strategies converge to similar error levels by 4096 steps. This is still useful, but the title and conclusion should be careful not to imply that language pretraining always gives a better forecaster.

---

### Official Review · Reviewer_sZec · 2026-05-20

**Rating:** 6
**Confidence:** 4

**Review:**

### Summary

This paper analyzes if language pretraining contributes to time series forecasting, beyond the architecture itself (i.e. transformer). The paper adapts a 0.6b qwen 3 for probabilistic forecasting by binning normalized time series values. Then, they train this architecture on gift-eval's training windows. They compare the language pretrained initialization against a randomly initialized architecture with full finetuning, finetuning of the embeddings and the model head (IO-only tuning), LoRA and LoRA+IO. The paper suggests that language pretrained model improves much earlier than random initialized models, especially for LoRA based fine-tuning.

### Strengths
1. I find the main finding of language pretraining giving a strong prior for time series forecasting, especially early on in training to be interesting. The main observation here is that pretrained attention LoRA reaches strong early performance relative to its random-initialized counterpart, while full finetuning and random initialization eventually become more comparable.
2. The paper attempts to explain why transfer occurs through frozen probes, retrieval from states, effective rank, and representation geometry etc. The main discussion presented in the paper for this is beyond my expertise to rigorously evaluate but I find the frozen-state and retrieval experiments especially interesting because they try to show that forecasting-relevant structure exists before any time series supervision.
3. The paper is openly and transparently discussing its limitations of using only one backbone architecture and one tokenization scheme.

### Areas for Improvement
1. Although interesting, the community interest now mostly in pretrained TSFMs such as Chronos, TimesFM and Moirai and these models has shown substantially stronger performance than LLM-pretrained models. I believe many of the techniques discussed in the second half of the paper (explanations about why transfer occurs) could be useful to analyze TSFMs and their adaptation through fine-tuning. I believe extending the scope of the paper to this setting would make it more relevant to the current community interests.
2. Although the paper is open about its limitations of using one model with one tokenization setting, deriving conclusions about overall language model training require experiments on diverse set of model architectures and also different tokenization variations. Also, to my understanding, the experiments are conducted with one seed. For training, different seeds could give different training curves. As the paper's main claim is that language pretraining gives benefits early on in adaptation, showing this with multiple seeds and with a confidence interval is important to derive conclusions.
3. To my understanding both initialization strategies (random and language pretrained) converge to similar error levels by 4096 steps. It is unclear to me if language pretraining gives any benefits in the end.
4. How does language pretraining compare with time series pretraining (for example with synthetic data)? This is an interesting and also a central question to justify why to use a language pretrained model.

### Detailed Comments
I understand that 4 page is limiting to present the figures but many figures are too difficult to read even when I zoomed them significantly. This is a minor polishing advice. Other than this, addressing the points 2 and 3 above would strengthen the paper significantly. Other than this, a definition and explanation of methods used in the paper would be helpful (for example you can explicitly state what IO stands for).

### Justification of Score

Although interesting, one architecture with one tokenization experiments, together with one seed results refrain me from advocating strongly about this paper. However, I believe the paper is interesting to the community.

---

### Official Review · Reviewer_t4XS · 2026-05-22
**Review for Submission 189**

**Rating:** 7
**Confidence:** 4

**Review:**

Review:

The paper investigates what language pretraining contributes to structured time-series forecasting beyond using a generic Transformer architecture. The authors adapt Qwen3-0.6B to probabilistic forecasting on GiftEval and compare language-pretrained models with identical randomly initialized models under full finetuning and parameter-efficient finetuning. The results show that language pretraining provides a strong early optimization advantage, especially under limited adaptation. The paper further supports this claim through LoRA experiments, frozen-state probes, retrieval-based forecasts, gradient coherence analysis, and effective-rank dynamics, arguing that language models contain reusable sequential structure that can be specialized for forecasting.

Strengths:

(1) The paper addresses an important question for structured-data foundation models: whether language-pretrained backbones provide useful transferable structure for time-series forecasting, or whether the benefit mainly comes from Transformer architecture.

(2) The comparison between language-pretrained Qwen3-0.6B and an identical randomly initialized model is well designed. This helps isolate the effect of pretraining from architecture and model size.

(3) The empirical findings are interesting and practically relevant. Language-pretrained models improve much faster in the early training regime, and attention-only LoRA recovers much of the transfer benefit of full finetuning.

(4) The paper provides several complementary analyses, including frozen linear probes, retrieval forecasts, gradient coherence, and effective-rank dynamics. These make the argument stronger than a standard forecasting benchmark comparison.

Areas for Improvement:

(1) The experimental scope is limited. The main results use one backbone, Qwen3-0.6B, one main benchmark, GiftEval, and univariate forecasting. More backbones, datasets, and multivariate settings would strengthen the generality of the claims.

(2) The paper should more clearly distinguish faster adaptation from better final forecasting performance. The results suggest that pretrained and randomly initialized models may converge to similar performance after enough training.

(3) Some representation analyses are suggestive but not fully conclusive. Frozen probes, retrieval forecasts, and cross-domain feature analysis indicate compatibility between language and time-series representations, but they do not fully prove the mechanism behind the finetuning gains.

(4) The method relies heavily on discretization and tokenization choices. More sensitivity analysis over vocabulary size, binning strategy, context length, and forecast horizon would improve robustness.

Detailed Comments:

(1) Please define “sequential prior” more explicitly. It is not fully clear whether this refers to optimization behavior, hidden-state geometry, attention structure, or all of these.

(2) Please include variance across multiple random seeds for the main transfer results, especially Table 1 and the early-training comparisons.

(3) The retrieval forecast result is interesting, but the win rate is only 37%. Please discuss which types of time-series patterns benefit from retrieval and which types fail.

(4) Please add ablations comparing LoRA on attention, MLP, and both. Since the paper emphasizes attention-only LoRA, this would help justify the claim that attention projections are the key low-rank adaptation path.